# The Wnt signaling receptor Fzd9 is essential for Myc-driven tumorigenesis in pancreatic islets

Mariano F Zacarías-Fluck[1],*, Toni Jauset[1,2],*, Sandra Martínez-Martín[1], Jastrinjan Kaur[1], Sílvia Casacuberta-Serra[2], Daniel Massó-Vallés[1], Erika Serrano del Pozo[1], Génesis Martín-Fernández[1], Íñigo González-Larreategui[1], Sergio López-Estévez[2], Lamorna Brown-Swigart[3], Marie-Eve Beaulieu[1], Jonathan R Whitfield[1], Babita Madan[4], David M Virshup[4], Gerard I Evan[5], Laura Soucek[1,2,6,7]

The huge cadre of genes regulated by Myc has obstructed the identification of critical effectors that are essential for Myc-driven tumorigenesis. Here, we describe how only the lack of the receptor Fzd9, previously identified as a Myc transcriptional target, impairs sustained tumor expansion and β-cell dedifferentiation in a mouse model of Myc-driven insulinoma, allows pancreatic islets to maintain their physiological structure and affects Myc-related global gene expression. Importantly, Wnt signaling inhibition in Fzd9-competent mice largely recapitulates the suppression of proliferation caused by Fzd9 deficiency upon Myc activation. Together, our results indicate that the Wnt signaling receptor Fzd9 is essential for Myc-induced tumorigenesis in pancreatic islets.

## Introduction

Myc is a highly pleiotropic transcription factor that governs cell expansion by coordinating diverse cellular processes, including proliferation, dedifferentiation, biosynthetic metabolism, cell growth, and angiogenesis (Dang, 2012). Using the Myc-switchable *pIns-MycER*[TAM]*;RIP7-Bcl-x*$_L$ β-cell tumor model in an unbiased reversible kinetic expression analysis, we previously identified genes required for maintenance of Myc-driven β-cell tumors. We found a relatively small subset of Myc target genes whose change in expression (whether induction or repression) was dependent upon sustained Myc activity, and that such changes were reversed by deactivation of Myc and consequent tumor regression. One of these candidate tumor maintenance genes was *Fzd9*, a member of the "frizzled" gene family of Wnt receptors (Lawlor et al, 2006). *Fzd9* was significantly

induced within 4 h of acute Myc activation in vivo, its expression was maintained as long as Myc activity was sustained, and rapidly decreased following Myc deactivation and tumor regression (Lawlor et al, 2006). In addition, *Fzd9* has been identified as a direct Myc target gene through expression and ChIP analysis (Lawlor et al, 2006; Sabò et al, 2014).

*Fzd9* is up-regulated in several types of human cancers including human gastric cancer (Kirikoshi et al, 2001), osteosarcoma (Wang et al, 2017) and astrocytoma (Zhang et al, 2006). Knockdown of *Fzd9* has also been shown to inhibit cell proliferation and motility in hepatocellular carcinoma cell lines (Fujimoto et al, 2009). However, as Fzd9 has also shown some tumor suppressor activity in acute myeloid leukemia (Zhang et al, 2016) and non-small cell lung cancer, its role in tumorigenesis is still controversial (Winn et al, 2005, 2006).

In this study, we explored the role of Fzd9 in mediating and maintaining Myc oncogenic function in *pIns-MycER*[TAM]*;RIP7-Bcl-x*$_L$ β-cell tumors. We show that Fzd9 does, indeed, play a critical function in the development of Myc-driven insulinomas. We, thus, identify a novel link connecting Myc and the Wnt pathway, which appears to be a required effector of Myc oncogenic activity in β-cell tumorigenesis. This implies the existence of a positive feedback loop, where Wnt/β-catenin signaling activates Myc (He et al, 1998), and Myc in turn activates Wnt/β-catenin signaling through Fzd9.

## Results

### Fzd9 is required for Myc-induced β-cell neoplasia

To determine whether Fzd9 plays a significant role in Myc-induced tumorigenesis in vivo, *pIns-MycER*[TAM]*;RIP7-Bcl-x*$_L$ (*MycER;BclXL* hereinafter) mice were crossed into a Fzd9-deficient background (Ranheim et al,

[1]Mouse Models of Cancer Therapy Group, Vall d'Hebron Institute of Oncology (VHIO), Vall d'Hebron Barcelona Hospital Campus, Barcelona, Spain   [2]Peptomyc SL, Vall d'Hebron Barcelona Hospital Campus, Barcelona, Spain   [3]Department of Pathology and Helen Diller Family Comprehensive Cancer Center, University of California, San Francisco, San Francisco, CA, USA   [4]Program in Cancer and Stem Cell Biology, Duke-National University of Singapore (NUS) Medical School, Singapore, Singapore   [5]Department of Biochemistry, University of Cambridge, Cambridge, UK   [6]Institució Catalana de Recerca i Estudis Avançats (ICREA), Barcelona, Spain   [7]Department of Biochemistry and Molecular Biology, Universitat Autònoma de Barcelona, Bellaterra, Spain

Correspondence: lsoucek@vhio.net
Daniel Massó-Vallés and Marie-Eve Beaulieu's present address is Peptomyc SL, Vall d'Hebron Barcelona Hospital Campus, Barcelona, Spain
*Mariano F Zacarías-Fluck and Toni Jauset contributed equally to this work

2005). The size, distribution, number and histological disposition of pancreatic islets in *Fzd9*$^{KO/KO}$ mice appeared identical to that in *Fzd9*$^{WT/WT}$ mice (Fig S1). When tamoxifen was systemically administered to *Fzd9*$^{WT/WT}$; *MycER;BclXL* and *Fzd9*$^{KO/KO}$;*MycER;BclXL* animals for 3-wk to activate MycER, control *Fzd9*$^{WT/WT}$;*MycER;BclXL* transgenic mice rapidly developed grossly hyperplastic insulinomas (Fig 1A and C), whereas *Fzd9*$^{KO/KO}$; *MycER;BclXL* mice exhibited no detectable β-cell tumor hyperplasia and pancreatic islets preserved their normal size (Fig 1B and C).

To exclude the possibility that Fzd9 deficiency inhibits MycER expression or stimulates its degradation, we monitored the presence of MycER immunohistochemically in β-cells, with an anti-ER antibody. As expected, MycER expression was undetectable in control *MycER* transgene-negative islets of *Fzd9*$^{KO/KO}$;*BclXL* mice, but was confirmed to be consistently expressed in tamoxifen-treated *Fzd9*$^{WT/WT}$;*MycER;* *BclXL* islets and was even higher in *Fzd9*$^{KO/KO}$;*MycER;BclXL* islets (Fig 1D). Hence, MycER is readily detectable in *Fzd9*-deficient cells, and therefore lack of Fzd9 impairs Myc-induced pancreatic β-cell expansion by mechanisms unrelated to suppression of MycER expression.

### MycER is functionally active in the absence of Fzd9

Since Fzd9 deficiency blocks the appearance of Myc-driven hyperplastic insulinomas, we next investigated the possibility that MycER is simply no longer functional in tamoxifen-treated *Fzd9*$^{KO/KO}$;*MycER;BclXL* islets. To do this, we first ascertained whether activated MycER retains its ability to drive β-cell proliferation in the absence of Fzd9. After 3 d of acute tamoxifen treatment, BrdU was systemically administered to *Fzd9*$^{KO/KO}$; *BclXL, Fzd9*$^{WT/WT}$;*MycER;BclXL* and *Fzd9*$^{KO/KO}$;*MycER;BclXL* mice 3 h before euthanasia. Then, pancreata were collected and BrdU incorporation into β-cells analyzed by immunofluorescence (Fig 2A). Multiple BrdU-positive β-cells were detected in both *Fzd9* wild-type (*Fzd9*$^{WT/WT}$;*MycER;BclXL*) and knockout (*Fzd9*$^{KO/KO}$;*MycER;BclXL*) mice (21.6% ± 7.7% and 24.1% ± 16.4%, respectively), whereas BrdU incorporation was almost absent from control MycER transgene-negative islets (*Fzd9*$^{KO/KO}$;*BclXL*; 0.4% ± 0.5%) (Fig 2B). Thus, at least initially, Myc retains its capacity to drive β-cell proliferation even in the absence of *Fzd9*.

Next, to ascertain whether Myc retains its capacity to induce apoptosis in pancreatic β-cells in the absence of Fzd9, MycER was continuously activated for 3 d in the β-cells of *Fzd9*$^{WT/WT}$;*MycER* and *Fzd9*$^{KO/KO}$;*MycER* mice, without co-expression of BclXL. As seen for proliferation, shrinkage of islets resulting from β-cell apoptosis (previously shown in Pelengaris et al [2002]) was observed in both *Fzd9*-proficient and *Fzd9*-deficient islets (Fig 2C). Hence, Fzd9 is not required for Myc-induced β-cell apoptosis.

Finally, additional Myc-dependent phenotypes previously described (Soucek et al, 2007), such as recruitment of mast cells (Fig S2A) and induction of angiogenesis (Fig S2B) were also observed after 3 d of tamoxifen treatment in *Fzd9*$^{KO/KO}$;*MycER;BclXL*.

Together, these observations indicate that, in the absence of Fzd9, MycER is still functionally active upon tamoxifen treatment.

### Fzd9 absence impairs tumor expansion

If Myc retains its ability, at least initially, to drive β-cell proliferation, why do islets fail to expand in tamoxifen-treated *Fzd9*$^{KO/KO}$;*MycER;BclXL* mice? To investigate this, we asked whether Myc-induced β-cell proliferation is maintained long term in the absence of Fzd9. MycER was activated in

β-cells for 3 wk and BrdU was administered systemically 3 h before euthanasia. Pancreata were harvested and BrdU incorporation detected by immunofluorescence (Fig 2D). High levels of BrdU-positive β-cells were detected in *Fzd9*$^{WT/WT}$;*MycER;BclXL* mice (16.49% ± 5.4%), but, strikingly, low levels of BrdU-positive cells (2.25% ± 1.49%) were observed in the *Fzd9*$^{KO/KO}$;*MycER;BclXL* β-cells. These levels were similar to those observed in MycER transgene-negative islets (*Fzd9*$^{KO/KO}$;*BclXL*; 0.52% ± 0.37%) (Fig 2E) and in stark contrast to the higher levels observed after 3 d (Fig 2B).

Immunohistochemical staining for the proliferation marker Ki67 demonstrated that the proliferative arrest observed in *Fzd9*$^{KO/KO}$;*MycER;* *BclXL* islets is in fact a progressive phenomenon already evident at 1 wk of sustained Myc activation and essentially complete by 3 wk (Fig 2F).

These results suggest that even though proliferation is initially triggered upon Myc activation in *Fzd9*$^{KO/KO}$;*MycER;BclXL* cells, Myc's tumorigenic potential is lost over time due to the lack of Fzd9.

### Fzd9 is necessary for *MycER*–dependent dedifferentiation and transformation of β-cells

In *MycER;BclXL* mice, partial Myc-induced dedifferentiation of β-cells has been previously noted as one of the pleiotropic effects of oncogenic Myc, responsible for tumorigenesis and expansion of the islets of Langerhans (Pelengaris et al, 2002). To verify if such an effect was Fzd9-dependent, we looked at the expression of both insulin and glucagon (markers of β-cells and α-cells, respectively). As expected, we detected high insulin expression in control *MycER* transgene-negative islets (*Fzd9*$^{KO/KO}$;*BclXL*) (Fig 3A). In contrast, insulin levels were somewhat repressed upon 3-wk tamoxifen treatment of *Fzd9*$^{WT/WT}$;*MycER;BclXL* mice because of Myc-induced partial dedifferentiation of β-cells (Fig 3B) (Pelengaris et al, 2002). Intriguingly, however, high levels of insulin expression were maintained in islets from *Fzd9*$^{KO/KO}$;*MycER;BclXL* mice after tamoxifen treatment (Fig 3C and D), indicating blunting of Myc-induced β-cell dedifferentiation in the absence of Fzd9. Indeed, pockets of fully differentiated cells can also be appreciated by hematoxylin and eosin (H&E) staining, already after 2 wk of MycER activation in *Fzd9*$^{KO/KO}$; *MycER;BclXL* but not in *Fzd9*$^{WT/WT}$;*MycER;BclXL* (Fig S3).

Glucagon positive α-cells displayed their typical proportion (~10% of the total islet cellular content) and distribution at the islet periphery in Fzd9-deficient mice without the *MycER* transgene (*Fzd9*$^{KO/KO}$;*BclXL*) (Fig 3A). In contrast, in tamoxifen-treated *Fzd9*$^{WT/WT}$;*MycER;BclXL* mice, Myc caused the appearance of α-cells scattered throughout the whole islet (Fig 3B). Notably, Myc did not trigger this relocalization in the absence of Fzd9 (*Fzd9*$^{KO/KO}$;*MycER;BclXL* mice treated with tamoxifen), where the α-cells displayed a more physiological distribution, although in a higher proportion than in control islets (Fig 3C and E).

Overall, these results suggest that Fzd9 is required for Myc to induce profound and long-lasting dedifferentiation in β-cells, as part of its tumorigenic program.

### Gene expression analysis shows down-regulation of genes involved in β-cell survival and stress response in the absence of Fzd9

To identify potential transcriptional changes that could explain the anti-tumorigenic effect observed in the absence of Fzd9, *Fzd9*$^{WT/WT}$; *MycER;BclXL* and *Fzd9*$^{KO/KO}$;*MycER;BclXL* mice were treated with tamoxifen for 3 d and the islet RNA was used for microarray

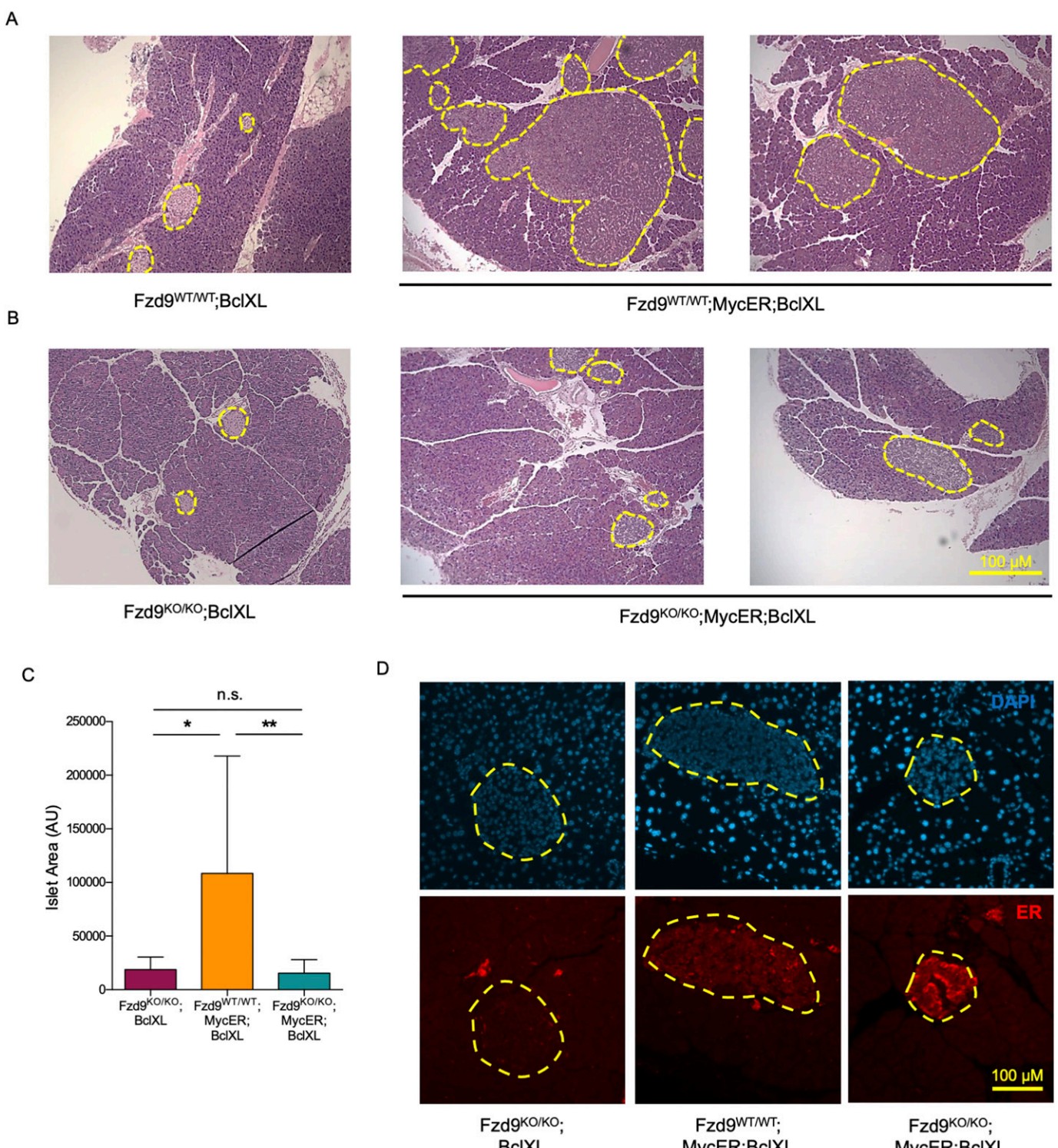

**Figure 1. The absence of Fzd9 impairs the development of Myc-driven pancreatic insulinomas.**
**(A, B)** H&E staining (4×) of pancreas sections from *Fzd9*-proficient and (B) *Fzd9*-deficient mice treated with tamoxifen for 3 wk. **(C)** Quantification of the islet size from *Fzd9^KO/KO;BclXL, Fzd9^WT/WT;MycER;BclXL* and *Fzd9^KO/KO;MycER;BclXL* mice. **(D)** Presence of MycER detected by immunofluorescence against ER (red) in these 3-wk treated islets. Yellow dotted lines define the periphery of the pancreatic islets. Representative images for each of the groups are shown (10×). Data information: in (C), data are represented as mean ± SD. * and ** indicate *P*-values below 0.05 and 0.01, respectively (Kruskal–Wallis followed by Dunn's test). **(A, D)** Scale bars: 200 *μ*m in (A), 100 *μ*m in (D).

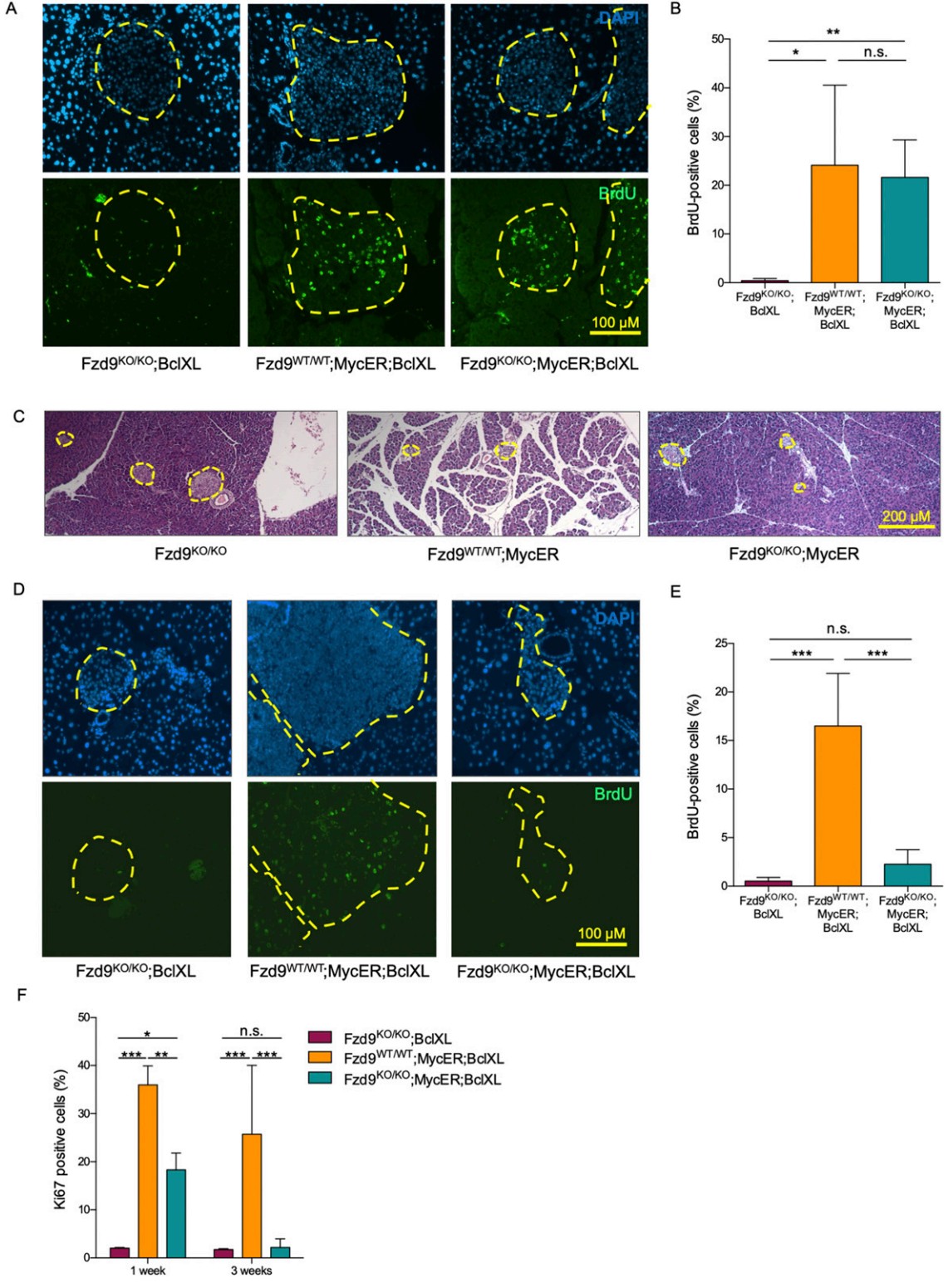

**Figure 2.  Sustained proliferation of pancreatic islets upon MycER activation requires Fzd9.**
**(A, B)** Incorporated BrdU (green) detected by immunofluorescence (10×) after 3 d of tamoxifen treatment and (B) its quantification. Percentages of BrdU-positive cells per islet from at least three mice per group are represented. **(C)** Representative H&E staining of pancreas sections from mice treated with tamoxifen for 3 d. Yellow dotted lines define the periphery of the pancreatic islets (4×). **(D, E)** Incorporated BrdU detected by immunofluorescence (10×) after 3 wk of tamoxifen treatment and (E) its quantification. Percentages of BrdU-positive cells per islet from at least three mice per group are represented. **(F)** Quantification of the proliferation marker Ki67 in islets from mice treated for 1 and 3 wk. Percentages of Ki67-positive cells per islet from three mice per group are represented. Data information: in (B, E, F), data are represented as mean ± SD. **(B, E, F)** Statistical significance of differences was examined using one-way, in (B, E), or two-way (in F) ANOVA followed by Tukey's test. ** and *** indicate $P$-values below 0.01 and 0.001, respectively. **(A, C, D)** Scale bars: 100 $\mu$m in (A, D), 200 $\mu$m in (C).

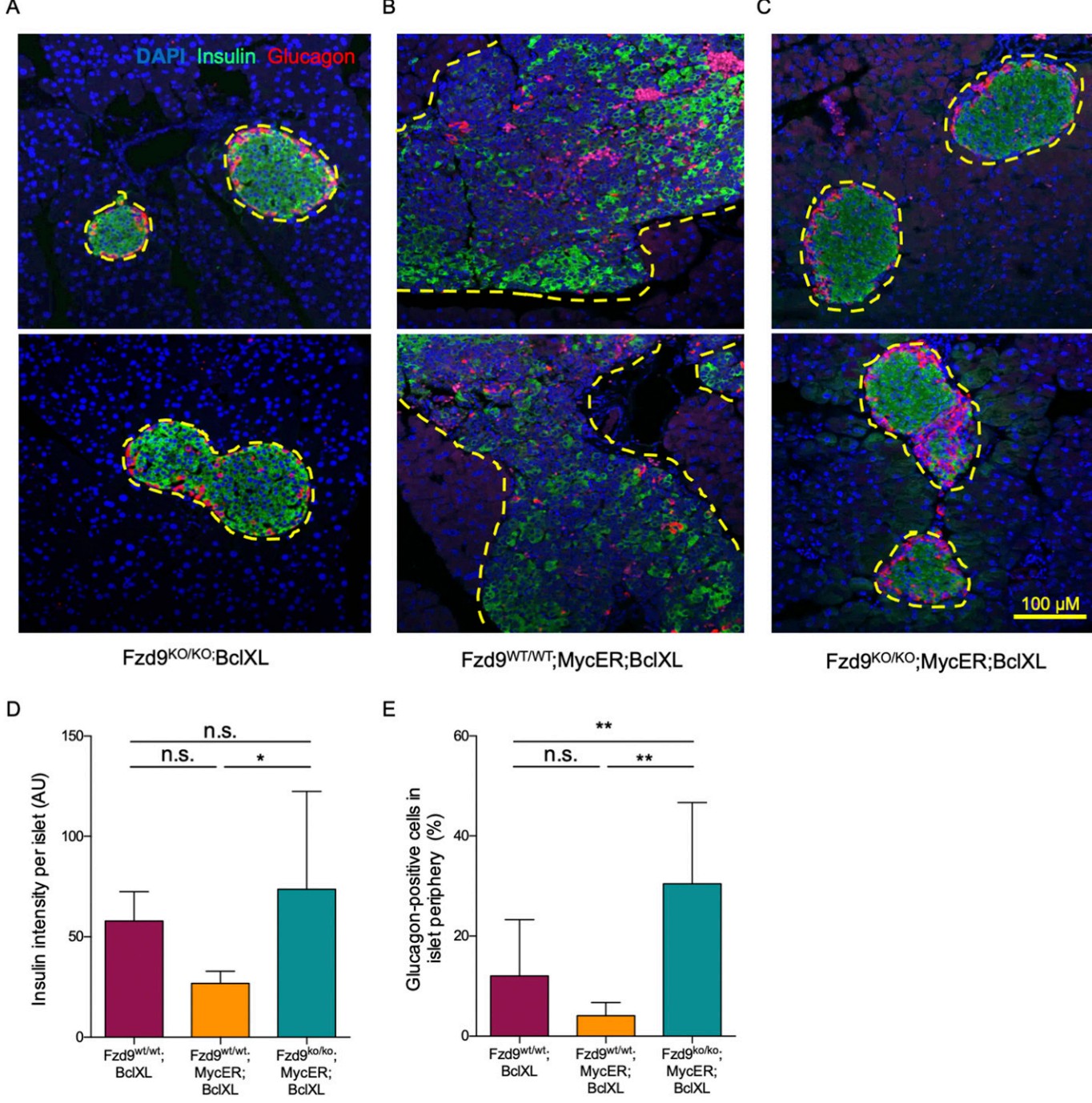

**Figure 3. Fzd9 mediates Myc-induced dedifferentiation of insulin-expressing β-cells.**
**(A, B, C)** Insulin (green) and glucagon (red) detected by immunofluorescence (10×) in pancreatic tissue sections of *Fzd9^{KO/KO};BclXL*, (B) *Fzd9^{WT/WT};MycER;BclXL*, and (C) *Fzd9^{KO/KO};MycER;BclXL* treated with tamoxifen for 3 wk. **(A, B, C, D)** Quantification of total insulin intensity in the islets from (A, B, C). **(E)** Quantification of glucagon positive cells in the periphery of the islets. Percentages of glucagon positive cells are shown. Data information: in (D, E), data are represented as mean ± SD. **(D, E)** Statistical significance of differences was examined using Kruskal–Wallis followed by Dunn's test (D) and one-way ANOVA followed by Tukey's test (E). * and ** indicate *P*-values below 0.05 and 0.01, respectively. Scale bar: 100 *µm*.

analysis. This early 3-d time-point was selected to reveal those differences in gene expression that could be the cause, and not the consequence, of the subsequent phenotypic changes (such as decrease proliferation rate or dedifferentiation of β-cells). Statistical analysis of the microarray (*P* < 0.05 and fold-change ± 1.2)

identified 933 differentially expressed genes out of 27,747 probes (Fig 4A and Table S1). Among the genes whose expression is most decreased in *Fzd9^{KO/KO}* pancreatic islets compared with *Fzd9^{WT/WT}* expression (fold-change < −2.5), we found several early response genes, comprising transcription factors and other cellular mediators

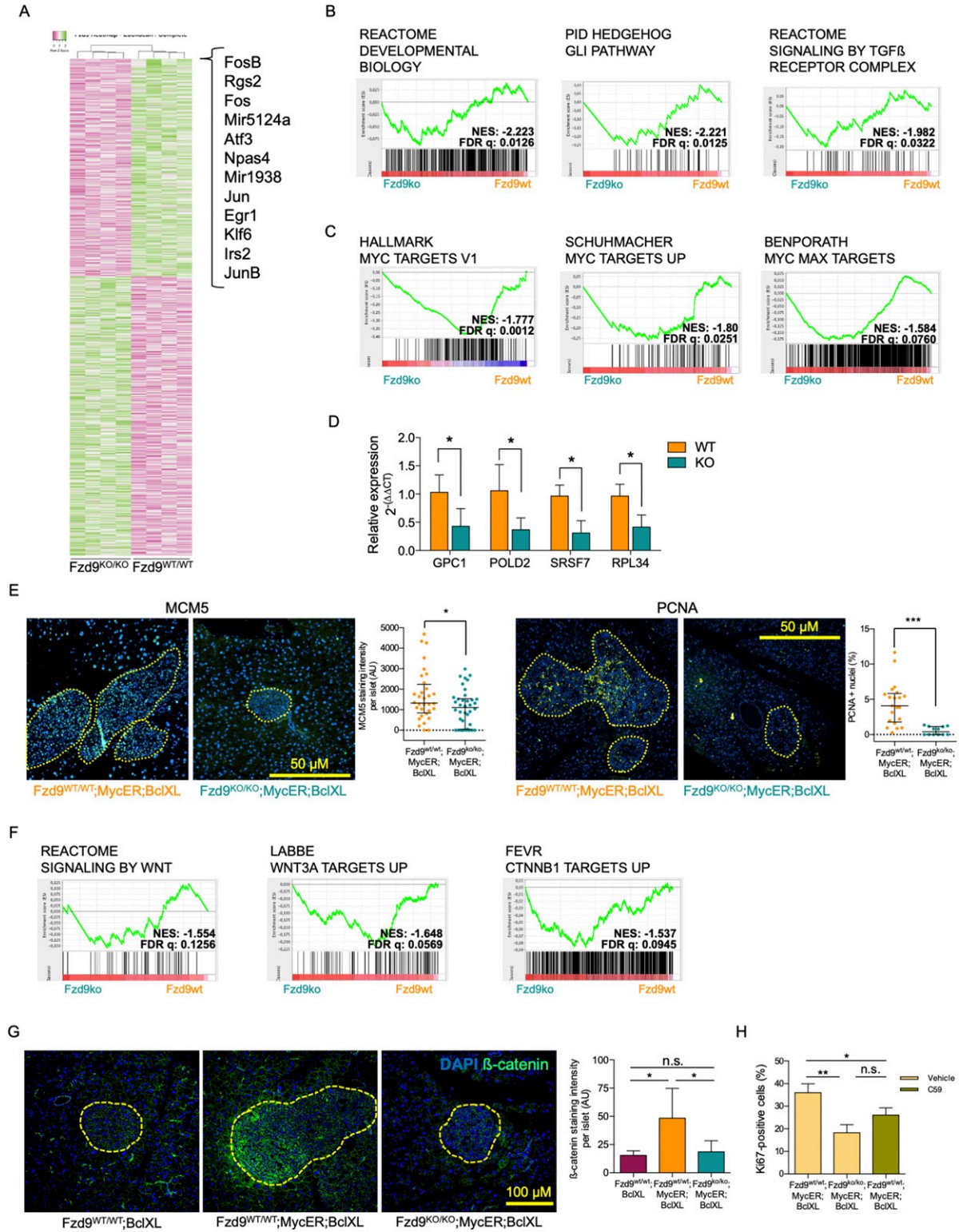

**Figure 4. The absence of Fzd9 alters the expression of Myc-related, differentiation and Wnt signaling Gene Sets upon MycER activation.**
**(A)** Heat map of differentially expressed genes determined by microarray analysis performed on a pool of multiple isolated islets (~20–40) from each of four *Fzd9^{WT/WT}; MycER;BclXL* mice and four *Fzd9^{KO/KO};MycER;BclXL* mice. **(B, C)** Gene sets related to pancreatic differentiation and (C) Myc targets. **(D)** qRT-PCR analysis of genes related to Gene Sets shown in (B) (GPC1) and in (C) (POLD2, SRSF7, and RPL34). **(E)** Immunofluorescence stainings (20×) and quantification of MYC targets MCM5 and PCNA. Total MCM5 intensity and PCNA positive nuclei per islet are shown. **(F)** Gene sets related to Wnt signaling pathway identified in the Gene Set Enrichment Analysis as differentially expressed when comparing expression profiles. **(G)** β-catenin detected by immunofluorescence (10×) in pancreatic tissue sections treated with tamoxifen for 3 d and its

of β-cell survival and stress response. *JUNB* and *ATF3*, for instance, are known to coordinate a β-cell survival pathway during inflammation (Gurzov et al, 2012), whereas *NPAS4* is an important early mediator of β-cell stress response (Sabatini et al, 2013). Moreover, *RGS2* (Dong et al, 2017) and *IRS2* (Blandino-Rosano et al, 2016) regulate β-cell survival and mass.

In addition, Gene Set Enrichment Analysis (GSEA) revealed significantly reduced mTORC1 signaling in *Fzd9^{KO/KO}* mice compared with *Fzd9^{WT/WT}* controls. Importantly, mTORC1 signaling is necessary for β-cell proliferation, as Raptor knockdown has a direct impact on β-cell size, mass, and survival (Blandino-Rosano et al, 2017) (NES: −2.18, q < 0.001; Table S2).

### Differentiation and Myc target gene sets are differentially expressed in the absence of Fzd9

GSEA showed that several gene sets associated to β-cell differentiation were differentially expressed in MycER expressing *Fzd9^{KO/KO}*;*MycER*;*BclXL* compared with *Fzd9^{WT/WT}* (Table S2). For instance, the expression of genes involved in developmental processes, including transcriptional regulation of pancreatic β-cell differentiation (https://reactome.org/content/detail/R-HSA-1266738) is significantly impaired (NES: −2.223; q: 0.0126). The Hedgehog/Gli pathway, whose up-regulation is responsible for β-cell dedifferentiation (Landsman et al, 2011), is significantly reduced (NES: −2.221, q: 0.0125), as is TGFβR signaling, also involved in β-cell dedifferentiation (Blum et al, 2014) (NES: −1.982, q: 0.0322) (Fig 4B). Together, these results show that several transcriptional programs engaged by Myc in β-cells, normally leading to their dedifferentiation, are significantly impaired in the absence of Fzd9.

Notably, GSEA also showed that several Myc-related gene sets are significantly down-regulated (Fig 4C and Table S3) along with Myc-related transcriptional programs like cell cycle, metabolism, and apoptosis (Fig S4A–C). The fact that the absence of only one Myc target gene is able to affect the expression of a number of Myc-related signatures reveals Fzd9 as a key effector of Myc-driven reprogramming of pancreatic β-cells.

We validated the transcriptional regulation in pancreatic islets observed in the GSEA by performing qRT-PCR of some of the identified differentially regulated genes after 3 d of tamoxifen-induced MycER activation in *Fzd9^{KO/KO}* and *Fzd9^{WT/WT}* mice (Fig 4D). We observed that, consistent with the microarray data, GPC1 (belonging to Developmental Biology Gene Set), RPL34, SRSF7 and POLD2 (MYC Targets Gene Set), were significantly down-regulated. This confirms—using a distinct quantitative method—the Fzd9-mediated transcriptional regulation of several MycER targets previously observed by microarray analysis.

We complemented this validation approach by performing immunofluorescence stainings for other down-regulated MYC target genes (namely, MCM5 and proliferating cell nuclear antigen [PCNA]) on pancreatic islets after 1 wk of MycER activation and confirmed their reduced expression in the islets of *Fzd9* knockout mice (Fig 4E).

### Wnt signaling is engaged by Fzd9 and acts as a downstream effector of Myc-induced tumorigenesis

Given the role of Fzd9 in Wnt signaling, this pathway seemed a priori a good candidate that could contribute to the multiple Myc-dependent phenotypes observed when MycER is activated. In fact, Wnt signaling has been shown to regulate pancreatic β-cell proliferation (Rulifson et al, 2007). However, as previously noted, there was no detectable difference in the histology of pancreata from *Fzd9^{KO/KO}* and *Fzd9^{WT/WT}* mice (Fig S1), indicating that the absence of this receptor is not rate limiting at least during normal development of the organ. Nevertheless, in MycER-activated islets, there are significant differences in Wnt-related gene sets between *Fzd9* knockout versus wild-type cells (*Fzd9^{KO/KO}*;*MycER*;*BclXL* versus *Fzd9^{WT/WT}*;*MycER*;*BclXL*) (Fig 4F). Moreover, qRT-PCR analysis of AXIN1 and JUNB, genes belonging to WNT and β-Catenin Gene Sets, respectively, confirmed their down-regulation, although they did not reach statistical significance (*P* = 0.11 and 0.099, respectively; Fig S4E). In addition, immunofluorescence staining for β-Actin, part of Labbe WNT3A Targets up Gene Set (Fig 4F), showed that in the absence of Fzd9 expression, β-Actin levels are significantly lower in pancreatic islets (Fig S4D).

Similarly, immunofluorescence against β-catenin revealed significantly increased protein levels of this Wnt signaling downstream effector in *Fzd9^{WT/WT}*;*MycER*;*BclXL* islets compared with their MycER-deficient counterpart *Fzd9^{WT/WT}*;*BclXL* (Fig 4G), whereas β-catenin staining remained low in *Fzd9^{KO/KO}*;*MycER*;*BclXL* cells. Hence, the Wnt signaling pathway, clearly engaged when Myc is active, appears dramatically affected by the absence of Fzd9.

Given these results and the well-established role of Frizzled receptors in Wnt signaling, we wondered whether pharmacological Wnt inhibition would be capable of recapitulating the Fzd9 tumor-resistant phenotype. To verify this hypothesis, we made use of C59, a potent inhibitor of *porcupine* (PORCN) that blocks Wnt palmitoylation, Wnt secretion and biological activity (Proffitt et al, 2013). *Fzd9^{WT/WT}*;*MycER*;*BclXL* mice were pretreated with C59 for 2 d, followed by 1 wk of C59/tamoxifen co-treatment. Then, pancreata from these C59-treated *Fzd9^{WT/WT}*;*MycER*;*BclXL* mice, as well as mice treated with tamoxifen alone (*Fzd9^{KO/KO}*;*BclXL*, *Fzd9^{WT/WT}*;*MycER*;*BclXL* and *Fzd9^{KO/KO}*;*MycER*;*BclXL*) were collected and stained to assess Ki67 positivity. Analysis of individual islets shows that, as described above (Fig 3D), when MycER is activated, cell proliferation in *Fzd9^{KO/KO}*;*MycER*;*BclXL* islets was significantly lower when compared with their *Fzd9^{WT/WT}*;*MycER*;*BclXL* counterparts (18.28% ± 3.5% versus 35.99% ± 3.9%). Notably, when the Fzd9-proficient mice were treated with C59, the proliferation rate was also significantly decreased (26.05% ± 3.2%) (Fig 4H), indicating that Wnt inhibition by C59 is able to largely mimic the phenotype of the Fzd9-deficient mice.

## Discussion

A vast amount of data in the literature points at Wnt signaling as one of the major culprits in solid and liquid tumors (Zhan et al, 2017) and Wnt

---

quantification in the islets. Total β-catenin intensity per islet area is shown. **(H)** Percentage of Ki67-positive cells in individual islets from *Fzd9^{WT/WT}*;*MycER*;*BclXL and Fzd9^{KO/KO}*;*MycER*;*BclXL* after 1 wk of tamoxifen treatment. An additional group of the *Fzd9^{WT/WT}*;*MycER*;*BclXL* mice was pretreated for 2 d with the Wnt inhibitor C59 and then received both C59 + tamoxifen during 1 wk. Data information: in (D, G, H), data are represented as mean ± SD, whereas in (E), as median and interquartile range. **(D, E, G)** Statistical significance of differences was examined using *t* test (D), Mann-Whitney U test (E) and Tukey's test (G). *, ** and *** indicate *P*-values below 0.05, 0.01 and 0.0001, respectively. **(E, G)** Scale bars: 50 μm in (E), 100 μm in (G). FDR, false discovery rate; NES, normalized enrichment score.

pathway inhibition via the targeting of Frizzled receptors has been suggested as a potential strategy to decrease growth and tumorigenicity of human tumors (Gurney et al, 2012). Our data indicate that simply inhibiting one receptor at a time (namely Fzd9) could be sufficient to achieve a significant therapeutic impact at least in some tumorigenic contexts, with the advantage of reducing potential side effects associated with the simultaneous inhibition of multiple Frizzled family members. Whereas Myc expression has been traditionally placed downstream of the Wnt signaling pathway, as previously discussed, others have indicated otherwise (Cowling & Cole, 2007). Here, we demonstrate for the first time the key role of the Wnt receptor Fzd9 and Wnt signaling in Myc-induced insulinomas. In this context, whereas previous results suggest that targeting Fzd9 may not be the best strategy for cancer therapy because of its dual pro- and anti-tumorigenic character (Zeng et al, 2018), our data indicate that it might be at least a valuable target when up-regulated in Myc-driven malignancies. In addition, our results suggest the existence of a novel positive feedback loop between Myc and Wnt signaling: deregulation of Myc might enhance Wnt signaling by up-regulation of Fzd9, which would, at the same time, promote Myc overexpression/deregulation—because Myc is a bona fide downstream target of Wnt—thus feeding again Wnt signaling through Fzd9. It should be noted that both our genetic and pharmacological approach to delete Fzd9 and interfere with Wnt signaling, respectively, are systemic and not specific for the islets of Langerhans only. Hence, we cannot exclude that such a positive feedback loop between Myc and Wnt can also have cell autonomous implications.

The identification of Fzd9 as a key Myc-driven tumorigenic effector is particularly relevant in view of the fact that, even if Myc is one of the "most wanted" cancer targets, several direct, and indirect Myc inhibition strategies have failed because of lack of efficacy and high toxicity derived from low specificity (Whitfield et al, 2017). Thus, identification of key tumorigenic effectors and their selective targeting represents an alternative strategy to achieve high therapeutic impact in Myc-deregulated malignancies. Here, we have described a good example of this alternative approach.

# Materials and Methods

## Generation and maintenance of genetically engineered mice

*pIns-MycER$^{TAM}$;RIP7-Bcl-x$_L$* and *Fzd9$^{KO/KO}$* mice have been previously described (Pelengaris et al, 2002; Zhao et al, 2005). All the animal studies were performed in accordance with the ARRIVE guidelines and the 3 Rs rule of Replacement, Reduction and Refinement principles. Animals were maintained and treated in accordance with protocols approved by the CEEA (Ethical Committee for the Use of Experimental Animals) at the Vall d'Hebron Institute of Oncology. Mice (both males and females) between 8 and 12 wk old were used.

## Preparation and administration of tamoxifen and C59

Tamoxifen (Sigma-Aldrich) was dissolved in peanut oil (Thermo Fisher Scientific) at 10 mg/ml. Aliquots of 1 ml were prepared and

frozen at −20°C. This injectable solution was administered to mice by intraperitoneal injection (6 $\mu$l/g) every 24 h. 1 ml-syringes and 27 G needles were used for injection.

The Wnt inhibitor C59 was formulated in PEG400 (Sigma-Aldrich) at 2 g/l. C59 in PEG400 was aliquoted and stored at 4°C. Right before each treatment, an equal amount of water (1:1) was added to make a final concentration of 1 g/l of C59 in 50% PEG400. Mice were then treated with a daily dose of 10 mg/kg by oral gavage for seven consecutive days. A mixture 1:1 dilution of water and PEG400 was used as vehicle for C59-untreated animals.

## Immunostaining of pancreas sections

18 h after the last administration of tamoxifen, mice were euthanized with $CO_2$ and pancreata collected. For BrdU staining, 150 $\mu$l of BrdU (Sigma-Aldrich) at 5 mg/ml were intraperitoneally injected 2 h before euthanasia.

For histological analysis, mouse pancreata were fixed with paraformaldehyde through systemic cardiac perfusion, collected in cassettes and further incubated in paraformaldehyde for 24 h. Tissues were then paraffin-embedded, and 5-$\mu$m sections cut and stained by H&E. Additional sections were used to perform immunostaining. In short, sections were deparaffinized, rehydrated, and subjected to high-temperature antigen retrieval in 10 mM citrate buffer (pH 6.0). Primary antibodies were as follows: anti-BrdU (Clone BU1/75; Bio-Rad), anti-Ki67 (SP6; Abcam), anti-Meca32 (Meca32; BD Biosciences), anti-insulin (EPR17359; Abcam), anti-glucagon (K79bB10; Abcam), anti-$\beta$-catenin (D10A8; Cell Signaling), anti-MCM5 (ab75975; Abcam), anti-PCNA (307904; BioLegend), and anti-$\beta$-Actin (A-5441; Sigma-Aldrich). Samples were incubated with primary antibodies overnight in blocking buffer (2.5% BSA, 0.3% Triton X-100 in PBS), sections were washed, and species-appropriate secondary applied, either Alexa Fluor 488 dye–conjugated antibodies (Thermo Fisher Scientific) or Vectastain ABC kit and DAB reagents (Vector Laboratories). Fluorescence antibody-labeled slides were mounted in DAKO fluorescent mounting medium containing 1 $\mu$g/ml DAPI counterstain.

## Microarray analysis of pancreatic islets

Genome-wide expression analysis was performed in isolated pancreatic islets. Briefly, tamoxifen-treated *Fzd9$^{WT/WT}$;MycER;BclXL* and *Fzd9$^{KO/KO}$;MycER;BclXL* (n = 4) mice were euthanized by cervical dislocation. Pancreata were inflated with Collagenase P (6 ml/mouse at 0.7 mg/ml) (Roche) in HBSS injected through the bile duct. Tissues were transferred to vials containing 5 ml of Collagenase P and incubated for 20 min at 37°C with gentle shaking. Digested pancreata were poured into 50 ml tubes and washed with cold HBSS by filling the tube, performing a short spin up to 652*g* and removing the supernatant. Pellets were resuspended in 5 ml and exocrine tissue further removed by filtering the suspension through 100 $\mu$m restrainers. Then, tissue remaining in the filter was placed in a 6 cm plate in 4 ml of cold HBSS. Pancreatic islets were visualized with the help of a magnifier by addition of dithizone at 0.02 g/l, hand-picked and transferred into a clean 1.5 ml tube.

RNA from islets was isolated, DNAse-treated and quality assessed through Agilent 2100 Bioanalyzer. RNA was reverse-transcribed to generate cDNA. Microarray was performed using a Mouse Gene Array 2.1 ST (Affimetryx). GSEA was performed using publicly available software provided by the Broad Institute (version 3.0) with the Hallmarks, Curated, Motif, gene ontology, Oncogenic Signatures and Immunological Signatures gene sets from the MsigDB (Subramanian et al, 2005). The number of permutations was set to 1,000 and the genes were ranked according to Signal2Noise. Heat map was generated using Heat mapper (Babicki et al, 2016). Hierarchical clustering was performed applying complete linkage method and based on Euclidean distance.

### qRT-PCR validation of microarray

qRT-PCR validation was performed in isolated pancreatic islets. Briefly, $Fzd9^{WT/WT}$;MycER;BclXL (n = 3) and $Fzd9^{KO/KO}$;MycER;BclXL (n = 3) were treated with tamoxifen for 3 d and pancreatic islets were then isolated from them following the protocol described in the previous section. RNA was then extracted from the samples using RNeasy kit (QIAGEN) and quantified using NanoDrop. Equal amounts of RNA were then DNAse-treated (NEB) and reverse transcribed to generate cDNA using iScript Reverse Transcription Supermix for RT-qPCR (Bio-Rad). SYBR green qRT-qPCR analysis was then performed on these cDNA samples with PerfeCTa SYBR Green FastMix, Low Rox (Quantabio) using QuantStudio 6 FLEX system (Applied Biosystems). The data thus obtained were analyzed following the comparative ($\Delta\Delta CT$) method described in Livak and Schmittgen (2001). $B2M$ (Beta-2-microglobulin) was used as the housekeeping gene. Based on our hypothesis and the data from the microarray, statistical analysis was performed using one-tailed $t$ test. Sequences of primers used are listed in Table S4.

### Image and statistical analysis

Islet size was quantified using ImageJ. Immunofluorescence (BrdU, Insulin, Glucagon, $\beta$-catenin, MCM5, PCNA, $\beta$-Actin) and Immunohistochemistry (Ki67) stainings were quantified using ImageJ and Qupath (Bankhead et al, 2017), respectively. All data were represented and analyzed using GraphPad Prism 6. One-way ANOVA and Tukey's test, or Kruskal–Wallis followed by Dunn's test were used to assess statistical significance when analyzing three groups of parametric or non-parametric distributions. Analysis of two groups were performed with $t$ test or Mann–Whitney. For each experiment, at least three animals per group were used and multiple islets quantified. Results are shown as mean ± SD or median and interquartile range, accordingly. Statistical significance was considered when $P < 0.05$.

## Data Availability

Microarray data were deposited under accession number GSE167073.

## Supplementary Information

## Acknowledgements

MF Zacarías-Fluck was supported by the Juan de la Cierva Programme of the Spanish Ministry of Economy and Competitiveness (IJCI-2014-22403) and L Soucek by the Generalitat de Catalunya AGAUR 2017 grant SGR-3193, S Martínez-Martín by the Generalitat de Catalunya "Contractació de personal investigador novell (FI-DGR)" 2016 fellowship (2016FI_B 00592). We thank the rest of the Soucek Lab for critical reading of the manuscript, Dr. Oriol Arqués for his advice on the microarray analysis and Dr. Águeda Martínez-Barriocanal for providing the proliferating cell nuclear antigen antibody. We also thank Dr. Jose Fernández-Navarro, Group Leader of Bioinformatics Unit at VHIO, for his advice on the statistical analysis of qRT-PCR. Finally, we thank the Cellex Foundation for providing research facilities and equipment and the FERO Foundation for their support to the laboratory of L Soucek.

## Author Contributions

MF Zacarias-Fluck: conceptualization, formal analysis, investigation, project administration, visualization, and writing—original draft, review, and editing.
T Jauset: conceptualization, formal analysis, investigation, project administration, visualization, and writing—original draft, review, and editing.
S Martínez-Martin: investigation.
J Kaur: investigation, formal analysis, and writing—review.
S Casacuberta-Serra: investigation.
D Massó-Vallés: investigation.
E Serrano del Pozo: investigation.
G Martín-Fernandez: investigation.
Í González-Larreategui: investigation.
S López-Estévez: investigation.
L Brown-Swigart: project administration.
M-E Beaulieu: formal analysis.
JR Whitfield: formal analysis and writing—original draft, review, and editing.
B Madan: resources.
DM Virshup: resources.
GI Evan: conceptualization and formal analysis.
L Soucek: conceptualization, formal analysis, project administration, supervision, investigation, and writing—original draft, review, and editing.

### Conflict of Interest Statement

Drs B Madan and DM Virshup have a financial interest in an unrelated PORCN inhibitor, ETC-159. The other authors declare no conflict of interest.

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
