## [Reviewer comments · Life Science Alliance]

Life Science Alliance

The Wnt signaling receptor Fzd9 is essential for Myc-driven tumorigenesis in pancreatic islets

Mariano Zacarias-Fluck, Toni Jauset, Sandra Martinez-Martin, Jastrinjan Kaur, Silvia Casacuberta-Serra, Daniel Masso-Valles, Erika Serrano del Pozo, Genesis Martin-Fernandez, Íñigo Gonzalez-Larreategui, Sergio López-Estévez, Lamorna Brown-Swigart, Marie-Eve Beaulieu, Jonathan Whitfield, Babita Madan, David Virshup, Gerard Evan, and Laura Soucek

DOI: <https://doi.org/10.26508/lsa.201900490>

Corresponding author(s): Laura Soucek, VHIO

Review Timeline:

Submission Date:	2019-07-18
Editorial Decision:	2019-08-08
Revision Received:	2021-01-25
Editorial Decision:	2021-02-17
Revision Received:	2021-02-22
Accepted:	2021-02-22

Scientific Editor: Shachi Bhatt

Transaction Report:

August 8, 2019

Re: Life Science Alliance manuscript #LSA-2019-00490

Dr. Laura Soucek
VHIO
Centre CELLEX
C/ Natzaret 115-117
Barcelona 08035
Spain

Dear Dr. Soucek,

Thank you for submitting your manuscript entitled "The Wnt signaling receptor Fzd9 is essential for Myc-driven tumorigenesis in pancreatic islets" to Life Science Alliance. The manuscript was assessed by expert reviewers, whose comments are appended to this letter.

As you will see, the reviewers appreciate your analyses and provide constructive input on how to further strengthen your work. I would thus like to invite you to submit a revised version of your manuscript to us, addressing the points raised by the reviewers. The requested control (rev#1, pt 1), validation (rev#1, pt 2), and quantification (rev#3, pt 1) need to get included, and it would be good to provide further physiological significance by following the suggestions of all three reviewers. This seems rather straightforward, but please do get in touch in case you would like to discuss an individual revision point further with me.

Thank you for this interesting contribution to Life Science Alliance. We are looking forward to receiving your revised manuscript.

Sincerely,

B. MANUSCRIPT ORGANIZATION AND FORMATTING:

Reviewer #1 (Comments to the Authors (Required)):

This data examines the role of Frizzled9(Fzd9), as a downstream effector of Myc signaling in driving tumorigenesis in pancreatic islets. The studies are largely based on genetic mouse models in which activation of Myc occurs in the presence or absence of Fzd9. The authors present data in which

deletion of Fzd9 inhibits islet tumor development, and perform gene expression analysis to begin to define pathways mediating these differences. These data are consistent with a model in which activation of Myc requires signaling through Fzd9 for sustained proliferation of islet cells, de-differentiation, and formation of tumors.

While the data are clearly presented, there are a number of issues that need to be addressed. These major points are listed below.

1. It is not clear which cell types express Fzd9 in this model. The mouse model used has a global deletion; therefore we cannot distinguish between loss of expression on islet cells or on other surrounding cell types. This could be addressed by either performing immunostaining, or if the antibodies are not reliable, in situ hybridization. This information is important in developing a better understanding of the pathways mediating the altered tumorigenesis: cell autonomous vs non-autonomous.
2. The gene expression profiling is of potential interest. However, there needs to be some validation of the predicted changes using either qRT-PCR or immunoblotting. The authors also need to provide some rationale for the choice of the timepoint at which the samples for RNA-seq were collected. In light of the data shown in Fig. 2 which indicates a progressive decrease in proliferation, some of the candidate genes could be examined at other time points.
3. The studies using the porcupine inhibitor C59 are very limited, and merely show decreased proliferation. At a minimum it would be important to know if this agent affects Myc activation and signaling as the authors have shown in Fig. 1 and supplemental data. Importantly, if this agent is acting through inhibiting Fzd9 it would be expected that the effects of C59 would be absent or at least diminished in mice with global Fzd9 deletion. This experiment would be relatively easy to perform, and would indicate that C59 is acting through inhibiting this receptor, rather than acting on other Wnt pathways.
4. The discussion at the end of the manuscript is very limited. The authors could speculate on the ramifications of this crosstalk between Myc and Wnt signaling, which is mentioned at the beginning of the paper. The data obtained in the study could be incorporated as supporting a feedback system.

Reviewer #2 (Comments to the Authors (Required)):

This paper provides evidence that FZD9 is required for Myc driven pancreatic tumorigenesis in a murine transgenic model. Because Myc is one of the first oncogenes to be discovered and an important oncogene in many different tumor types, it is inherently interesting to understand the mechanisms and downstream pathways involved in how Myc functions in tumor growth. Furthermore, Myc is a currently an "undruggable" target, so these data indicate a potential strategy for therapeutic intervention. Based on previous work, it should be feasible to develop a FZD9 selective antibody that would reduce Wnt signaling.

This paper could be improved by some straightforward analyses. It would be interesting to know if FZD9 and Myc expression are correlated in human pancreatic tumors (or any other human tumor type where Myc or FZD9 are over-expressed).

Reviewer #3 (Comments to the Authors (Required)):

Jauset et al. have interrogated the functional role of the Wnt Signaling receptor Fzd9 in MYC-driven tumorigenesis in pancreatic islets. The rationale for the focus on Fzd9 is sound and the work performed is up to the usual high standards of the Soucek lab. The manuscript is easy to read and the experiments performed are logical and well-controlled. Interpretation of results is also sound.

This work was performed in the appropriate mouse models, and thus the results hold weight as they are performed in vivo. I only have a few small points to suggest in an effort to further strengthen the manuscript.

1. The authors may wish to add quantification to Figures 3C and 4E.
2. Perhaps showing a figure highlighting that MYC and Fzd9 are co-expressed in human pancreatic cancer would show that these results have relevance to human disease. This would add further weight to their suggestion that inhibiting Fzd9 has merit as an approach to target MYC activity.
3. Remove 'though' in the sentence: "When tamoxifen was systemically administered to Fzd9^{WT/WT};MycER;BclXL and Fzd9^{KO/KO};MycER;BclXL animals for 3 weeks to activate MycER, though, control Fzd9^{WT/WT};MycER;BclX...."

We would like to thank the editor and the reviewers for taking the time to read our manuscript and for their constructive criticism and suggestions on how this manuscript could be further improved. We have attempted to address the reviewers' concerns to our best ability and hope that our responses below (and the corresponding changes to the manuscript) are up to the mark.

We have addressed the following concerns following the editor's recommendations:

Reviewer #1 (Comments to the Authors (Required)):

This data examines the role of Frizzled9(Fzd9), as a downstream effector of Myc signaling in driving tumorigenesis in pancreatic islets. The studies are largely based on genetic mouse models in which activation of Myc occurs in the presence or absence of Fzd9. The authors present data in which deletion of Fzd9 inhibits islet tumor development, and perform gene expression analysis to begin to define pathways mediating these differences. These data are consistent with a model in which activation of Myc requires signaling through Fzd9 for sustained proliferation of islet cells, differentiation, and formation of tumors. While the data are clearly presented, there are a number of issues that need to be addressed. These major points are listed below.

1. It is not clear which cell types express Fzd9 in this model. The mouse model used has a global deletion; therefore we cannot distinguish between loss of expression on islet cells or on other surrounding cell types. This could be addressed by either performing immunostaining, or if the antibodies are not reliable, in situ hybridization. This information is important in developing a better understanding of the pathways mediating the altered tumorigenesis: cell autonomous vs non-autonomous.

We thank the reviewer for his/her comment.

To address this concern, we used 2 previously published commercial anti-Fzd9 antibodies (Abcam ab61430; R&D systems AF2440) to perform IF/IHC in Fzd9 knock-out pancreatic tissue. Surprisingly, we found that both antibodies showed positive staining in the Fzd9 knock-out pancreatic tissue. On the other hand, qPCR analysis on islets derived from Fzd9 KO mice used showed that these islets did **not** express Fzd9, consistently with their genotype, pointing to the possibility of non-specific antibody staining. In addition, the fact that this mouse model was generated by removing the 1.8kb unique Fzd9 exon excludes the possibility of any truncated Fzd9 fragments or splice variants still being expressed and therefore detected by the antibodies in this model. Together, these data indicate that both antibodies gave a non-specific staining and we could therefore not address this comment through IHC. However, we would like to remind the reviewer that, as shown in a previous publication (Lawlor et al. 2006), this upregulation of Fzd9 upon MYC-ER activation takes place only in the pancreatic islets, since MYC-ER expression is limited to this tissue only.

Further, while we agree that determining whether the effect of Fzd9 is cell autonomous vs. non-autonomous is a biologically relevant question, we argue that, from a translational point of view, wherein interference with the Wnt pathway will take place at the systemic level (thereby mimicking a global Fzd9 KO or C59 treatment as shown in our manuscript), the point raised by the reviewer is less relevant in the context of our manuscript.

If, despite our efforts in clarifying this point, this experiment is deemed necessary for this manuscript, we will proceed with setting up in situ hybridization to answer this question.

2. The gene expression profiling is of potential interest. However, there needs to be some validation of the predicted changes using either qRT-PCR or immunoblotting. The authors also need to provide some rationale for the choice of the timepoint at which the samples for RNA-seq were collected. In light of the data shown in Fig. 2 which indicates a progressive decrease in proliferation, some or the candidate genes could be examined at other time points.

We appreciate the reviewer's suggestion and the chance to better explain the selection of this time-point. As described in Figure 2, and as highlighted by the reviewer, there is a progressive decrease of proliferation. In addition, as demonstrated in this study and also a previous study, MYC-ER activation causes dedifferentiation of pancreatic islets. This dedifferentiation is the consequence of changes in the gene expression profile of the cells within the islets. We anticipated that expression analysis at later time-points would reflect the gene expression profiles of differentiated vs. undifferentiated cells mainly, which is not the aim of this study. Instead, we are interested in investigating the differences in gene expression that will cause this dedifferentiation. Therefore, we selected 3 days, since at this time-point there is still no phenotypic change caused by MYC in the absence of Fzd9. Thus, the differences in gene expression at 3 days reveal those that result from MYC activation in the absence of Fzd9, and not those differences that derive from the downstream phenotypic changes resulting from this activation. A sentence summarizing this explanation has been added into the manuscript on page 6 and 7.

With regards to the validation of gene regulation observed by microarray analysis, we performed qPCR analysis on RNA obtained from MYC-ER-activated Fzd9 wt or KO mice. We selected six genes for this analysis, representing various differentially regulated gene sets identified in the gene set enrichment analysis (GSEA). Of these, 4 genes (POLD2, GPC1, RPL34 and SRSF7) were significantly downregulated while JUNB and β -actin showed a downregulation although without reaching statistical significance, possibly due to the small "n" of the study. Moreover, we performed immunofluorescence of pancreatic tissue and confirmed the downregulation of MCM5, PCNA and β -Actin in the islets of Fzd9 KO mice. These results confirm the microarray data demonstrating a downregulation of several genes across various gene sets upon Fzd9 KO in a Myc-activated background. This data has been added as Figure 4D-E and Supplementary Figure 4E in the new incarnation of our manuscript along with a textual description on Page 8.

3. The studies using the porcupine inhibitor C59 are very limited, and merely show decreased proliferation. At a minimum it would be important to know if this agent affects Myc activation and signaling as the authors have shown in Fig. 1 and supplemental data. Importantly, if this agent is acting through inhibiting Fzd9 it would be expected that the effects of C59 would be absent or at least diminished in mice with global Fzd9 deletion. This experiment would be relatively easy to perform, and would indicate that C59 is acting through inhibiting this receptor, rather than acting on other Wnt pathways.

(NOT DEEMED NECESSARY BY THE EDITOR)

4. The discussion at the end of the manuscript is very limited. The authors could speculate on the ramifications of this crosstalk between Myc and Wnt signaling, which is mentioned at the beginning of the paper. The data obtained in the study could be incorporated as supporting a feedback system.

We thank the reviewer for the suggestion. We have included a few sentences at the end of the manuscript to further discuss the implications of this observation.

Reviewer #2 (Comments to the Authors (Required)):

This paper provides evidence that FZD9 is required for Myc driven pancreatic tumorigenesis in a murine transgenic model. Because Myc is one of the first oncogenes to be discovered and an important oncogene in many different tumor types, it is inherently interesting to understand the mechanisms and downstream pathways involved in how Myc functions in tumor growth. Furthermore, Myc is a currently an "undruggable" target, so these data indicate a potential strategy for therapeutic intervention. Based on previous work, it should be feasible to develop a FZD9 selective antibody that would reduce Wnt signaling.

This paper could be improved by some straightforward analyses. It would be interesting to know if FZD9 and Myc expression are correlated in human pancreatic tumors (or any other human tumor type where Myc or FZD9 are over-expressed).

We appreciate the input of the reviewer and have performed this analysis using publically available mRNA expression data from Cancer Cell Line Encyclopedia (CCLE).

We performed correlation analysis between expression levels of MYC and Fzd9 in cell lines derived from different cancer subtypes including pancreatic, breast, ovarian, lung, CNS and colon cancers and melanoma (see below). While the reviewer mentions pancreatic cancer, we must point out that this insulinoma model does not represent the most prevalent and aggressive pancreatic ductal adenocarcinoma (PDAC). Nevertheless, below we have included a correlation analysis in pancreatic cancer as a whole.

Interestingly, the correlations showed that breast cancer was the only subtype of cancer cell lines that display a statistical significance between MYC and Fzd9 expression. Following this observation, we are now working on this research line in breast cancer and we hope to be able to publish the results in the future, and since the data is not directly relevant to insulinoma or pancreatic cancer, we have not included it in the present manuscript. We hope the reviewer will understand this choice.

While there is no significant correlation between Fzd9 and Myc expression in pancreatic cancers as a whole, we do not exclude the possibility that Fzd9 inhibition could be an effective therapeutic option for individual tumors presenting both MYC deregulation and Fzd9 overexpression.

Reviewer #3 (Comments to the Authors (Required)):

Jauset et al. have interrogated the functional role of the Wnt Signaling receptor Fzd9 in MYC-driven tumorigenesis in pancreatic islets. The rationale for the focus on Fzd9 is sound and the work performed is up to the usual high standards of the Soucek lab. The manuscript is easy to read and the experiments performed are logical and well-controlled. Interpretation of results is also sound. This work was performed in the appropriate mouse models, and thus the results hold weight as they are performed in vivo. I only have a few small points to suggest in an effort to further strengthen the manuscript.

1. The authors may wish to add quantification to Figures 3C and 4E.

Quantifications as Figure 3D-E and Figure 4G have been included following the reviewer's suggestion. The figures are also mentioned within the body of text when appropriate.

2. Perhaps showing a figure highlighting that MYC and Fzd9 are co-expressed in human pancreatic cancer would show that these results have relevance to human disease. This would add further weight to their suggestion that inhibiting Fzd9 has merit as an approach to target MYC activity.

We value the reviewer's input and have addressed this issue with Reviewer #2 (see above).

3. Remove 'though' in the sentence: "When tamoxifen was systemically administered to Fzd9WT/WT;MycER;BclXL and Fzd9KO/KO;MycER;BclXL animals for 3 weeks to activate MycER, though, control Fzd9WT/WT;MycER;BclX...."

We thank the reviewer for pointing this out. The word has been removed.

February 17, 2021

RE: Life Science Alliance Manuscript #LSA-2019-00490-TR

Dr. Laura Soucek
VHIO
Centre CELLEX
C/ Natzaret 115-117
Barcelona 08035
Spain

Dear Dr. Soucek,

Thank you for submitting your revised manuscript entitled "The Wnt signaling receptor Fzd9 is essential for Myc-driven tumorigenesis in pancreatic islets". We would be happy to publish your paper in Life Science Alliance pending final revisions necessary to meet our formatting guidelines.

The request for in situ hybridization data for Fzd9 (Reviewer 1, and pbp response) should be included in the revised manuscript, only if the data is readily attainable. If the data is not readily available or attainable, then the authors should discuss this concern.

Along with the points listed below, please also attend to the following:

- please consult our manuscript preparation guidelines <https://www.life-science-alliance.org/manuscript-prep> and make sure your manuscript sections are in the correct order;
- please separate the Results and Discussion section into two - 1. Results 2. Discussion, as per our formatting requirements
- please upload your main and supplementary figures as single files
- please move your main, supplementary figure and table legends in your ms text after the references section
- please upload your tables in editable .doc or .xls file
- please upload your main manuscript text as an editable doc file
- please provide scale bars for Figures 1A; 2C; 3A, B; 4E; S4D
- please deposit the microarray data into a publicly available database and provide the accession number in the revised manuscript

A. FINAL FILES:

B. MANUSCRIPT ORGANIZATION AND FORMATTING:

Sincerely,

Shachi Bhatt, Ph.D.

Executive Editor

Life Science Alliance

<https://www.lsjournal.org/>

Interested in an editorial career? EMBO Solutions is hiring a Scientific Editor to join the international Life Science Alliance team. Find out more here -

https://www.embo.org/documents/jobs/Vacancy_Notice_Scientific_editor_LSA.pdf

Reviewer #1 (Comments to the Authors (Required)):

This is a revised manuscript examining the role of Fzd9 in mediating the effects of Myc overexpression on pancreatic tumors. The studies have investigated this using genetic mouse models in which Myc is driven in the presence or absence of Fzd9 receptors. The data clearly show an important role for Fzd9 in mediating these effects. In addition, gene expression of pancreatic islets has been performed to identify changes that are dependent on Fzd9 expression. Finally a pharmacological inhibitor of Wnt signaling was employed that mimicked the effects of Fzd9 deletion.

Overall, the data supports the conclusions, and the authors have been responsive to the previous critiques. The only issue that was raised and has not been addressed are experiments to determine the cell type expressing Fzd9. The authors performed immunostaining with two Fzd9 antibodies and used appropriate controls. Unfortunately, these antibodies appeared to give non-specific staining. Nevertheless, this is an important issue, especially in light of the complex role for Fzd9, with both pro and anti-tumor effects being reported. The authors indicate a willingness to perform in situ hybridization, and this result would strengthen the paper.

Reviewer #2 (Comments to the Authors (Required)):

This work provides novel insight into the mechanisms of Myc-driven tumorigenesis in a transgenic mouse model by connecting Myc to Wnt signaling. The revised manuscript is acceptable for publication.

Reviewer #3 (Comments to the Authors (Required)):

The authors have addressed my concerns.

February 22, 2021

RE: Life Science Alliance Manuscript #LSA-2019-00490-TRR

Dr. Laura Soucek
VHIO
Centre CELLEX
C/ Natzaret 115-117
Barcelona 08035
Spain

Dear Dr. Soucek,

Thank you for submitting your Research Article entitled "The Wnt signaling receptor Fzd9 is essential for Myc-driven tumorigenesis in pancreatic islets". It is a pleasure to let you know that your manuscript is now accepted for publication in Life Science Alliance. Congratulations on this interesting work.

DISTRIBUTION OF MATERIALS:

Again, congratulations on a very nice paper. I hope you found the review process to be constructive and are pleased with how the manuscript was handled editorially. We look forward to future exciting submissions from your lab.

Sincerely,

Shachi Bhatt, Ph.D.

Executive Editor

Life Science Alliance

<https://www.lsjournal.org/>

Interested in an editorial career? EMBO Solutions is hiring a Scientific Editor to join the international Life Science Alliance team. Find out more here -

https://www.embo.org/documents/jobs/Vacancy_Notice_Scientific_editor_LSA.pdf